# All-sky AMSU-A radiance data assimilation using the gain-form of Local Ensemble Transform Kalman filter within MPAS-JEDI-2.1.0: implementation, tuning, and evaluation

Tao Sun¹, Jonathan J. Guerrette¹.\*, Zhiquan Liu¹, Junmei Ban¹, Byoung-Joo Jung¹, Ivette Hernandez Banos¹, and Chris Snyder¹

<sup>1</sup>NSF National Center for Atmospheric Research, Boulder, CO 80301, USA

\*now at: Tomorrow.io, Golden, CO 80401, USA

Correspondence to: Tao Sun (taosun@ucar.edu)

Abstract. The Gain-form of Local Ensemble Transform Kalman Filter (LGETKF) has been implemented in the Joint Effort for Data assimilation Integration (JEDI) with the Model for Prediction Across Scales - Atmosphere (MPAS-A) (i.e., MPAS-JEDI). LGETKF applies vertical localization in model space and is particularly convenient for assimilating satellite radiances that do not have an explicit vertical height assigned to each channel. Additional efforts are made to optimize the ensemble analysis procedure and improve the computational efficiency of MPAS-JEDI's LGETKF. This is the first application of JEDI-based LGETKF for assimilating radiance data in all-weather situations with a global MPAS configuration. The system is firstly tuned for covariance inflation and horizontal localization settings. It is found that using a combination of relaxation to prior perturbation (RTPP) and relaxation to prior spread (RTPS) outperforms using RTPP or RTPS alone, and using a smaller horizontal localization scale for all-sky radiances is preferable. With the optimized inflation and localization settings, assimilating all-sky radiances of the Advanced Microwave Sounding Unit - A (AMSU-A) window channels with an 80-member LGETKF improved the forecasts of moisture, wind, clouds, and precipitation fields, when compared to the benchmark experiment without assimilation of all-sky AMSU-A radiances. The positive forecast impact of all-sky AMSU-A radiances is the largest over the tropical regions up to 7-day. Some degradation on the temperature forecasts is seen over certain regions, where the model forecast is likely biased, causing deficiencies for assimilating all-sky data. The LGETKF capability is available in the recent public release of MPAS-JEDI and ready for research and operational explorations.

#### 1 Introduction

The Joint Effort for Data assimilation Integration (JEDI) is a new data assimilation (DA) framework with model-agnostic components that can be interfaced to multiple models (Trémolet and Auligné 2020). The development of JEDI is led by the Joint Center for Satellite Data Assimilation (JCSDA) with partners from several research and operational institutes. Considerable efforts at the National Center for Atmospheric Research (NCAR) have been devoted to the development of a JEDI-based DA system for the Model for Prediction Across Scales – Atmosphere (MPAS-A; Skamarock et al., 2012), which

we will call MPAS-JEDI. (For historical reasons, this system has been previously referred to as JEDI-MPAS, such as in Liu et al. 2022.) Variational DA approaches have been successfully implemented into MPAS-JEDI for deterministic analysis, including the three-dimensional variational (3DVar) method, the ensemble-variational (EnVar) method, and hybrid variants of EnVar (Liu et al., 2022; Jung et al., 2024).

Ensemble DA plays a vital role in numerical weather prediction (NWP) by providing initial conditions for ensemble forecasts, which can also be used as input for a deterministic EnVar analysis to form flow-dependent background error covariances (BEC). There are two general classes of ensemble DA methods, the ensemble of data assimilations (EDA; Houtekamer et al. 1996) and various versions of the Ensemble Kalman Filter (EnKF; Evensen 2003). For MPAS-JEDI, Guerrette et al. (2023) implemented EDA that performs an ensemble of 3DEnVar analysis with perturbed observations. They demonstrated a comparable or superior performance for EDA when compared to the Ensemble Adjustment Kalman Filter (EAKF; Anderson 2001) within the Data Assimilation of Research Testbed (DART; Anderson et al., 2009, Ha et al. 2017), but with a substantially larger computational cost than the EAKF. Therefore, it is worth implementing an EnKF-based ensemble DA capability for

MPAS-JEDI.

Frolov et al. (2024) implemented two EnKF-based methods within the JEDI framework, including the Local Ensemble Transform Kalman Filter (LETKF; Hunt et al. 2007) and the gain-form LETKF (LGETKF; Bishop et al. 2017). Frolov et al. (2024) provided a comprehensive description of both methods and demonstrated their successful application to different components of the Earth system model. Park et al. (2023) showed that the limited-area FV3-JEDI's LETKF and LGETKF have a comparable performance in the convective-scale radar DA when compared to the Ensemble Square Root Kalman Filter (EnSRF; Whitaker and Hamill 2002) in the Gridpoint Statistical Interpolation (GSI; Shao et al. 2016) DA system. Compared to the LETKF, the LGETKF has the advantage in performing vertical localization in model space (Bishop et al. 2017), making it particularly suitable for the assimilation of satellite radiances that are column-integrated measurements and are hard to define their vertical coordinates. Various studies have proven that the model space vertical localization generally outperforms observation space vertical localization (Campbell et al. 2010; Bishop et al. 2017; Lei et al. 2018). Given the feasibility of the

55 The cloud- and precipitation-affected satellite radiances from microwave (MW) instruments have been operationally assimilated in the global NWP models in recent decades (e.g., Bauer et al., 2010; Zhu et al., 2016; 2019; Geer et al., 2017, 2018; Migliorini and Candy, 2019; Shahabadi and Buehner, 2024) with the variational-based DA systems. EnKF-based methods were also employed to assimilate all-sky infrared radiances for case studies with regional models (e.g., Zhang et al., 2016; Martinet and Zhang, 2018; Hoda et al., 2018; Zhang et al., 2019; Okamoto et al, 2019; Chan et al., 2020; Zhu et al., 60 2022). Fewer studies, however, have explored all-sky assimilation of MW radiances using EnKF-based methods for global

LGETKF algorithm within the JEDI framework, implementing it to MPAS-JEDI is a worthwhile endeavour, as it better

handles the vertical localization of satellite radiance assimilation.

models. Bonavita et al. (2020) presented a comprehensive evaluation of the all-sky MW radiance assimilation with the European Centre for Medium-Range Weather Forecasts (ECMWF)'s EnKF system, and their results showed that EnKF could properly extract wind information from all-sky radiance observations and brings 2%-4% improvement in forecast scores.

Deleted: Therefore

Deleted: the

Deleted: LGETKF

Deleted: method is chosen in this study to

Deleted: e

In this study, the LGETKF algorithm within the JEDI framework is implemented in MPAS-JEDI, and its performance is evaluated through the clear-sky and all-sky MW radiance assimilation. As the first implementation of LGETKF in MPAS-JEDI, we first tune the assimilation configurations such as covariance inflation and localization, providing a reference for the community on achieving stable and robust performance. We then access the forecast impact of all-sky assimilation of AMSU-A radiances over a month-long period, offering deeper insight into the performance of MPAS-JEDI's LGETKF. The rest of the paper is organized as follows. In section 2, the implementation of the LGETKF in MPAS-JEDI is detailed. The MPAS 75 model configuration, all-sky radiance DA method, and experimental design are provided in section 3. The results of tuning the covariance inflation and horizontal localization are given in section 4, followed by our evaluation of all-sky AMSU-A radiance DA with the LGETKF in section 5 and conclusions in section 6.

2 Methodology

## 2.1 LGETKF in MPAS-JEDI

#### 80 2.1.1 Implementation of LGETKF in MPAS-JEDI

The local volume solvers developed by Frolov et al. (2024), which are part of the Object-Oriented Prediction System (OOPS), the DA solver component of JEDI, have been implemented in MPAS-JEDI. Both the LETKF and LGETKF are now available in MPAS-JEDI, with the latter being the focus of this study. In the LGETKF, the model states within a local volume (i.e., a vertical column of a horizontal grid point) are updated using the surrounding observations, and the updates in each volume occur simultaneously and independently from other volumes. When computing the Kalman gain, the LGETKF employs perturbations from an expanded ensemble, which is obtained by modulating the original ensemble with the eigenvectors of the vertical localization matrix (Bishop et al. 2017). Thus, the LGETKF has the advantage of performing vertical localization in model space and is particularly suitable for assimilating non-local observations such as satellite radiances (Lei et al. 2018). For a detailed description of the LGETKF algorithm within OOPS, see Frolov et al. (2024).

The analysis variables of MPAS-JEDI's LGETKF include temperature (T), horizontal wind components (U, V), surface pressure (P<sub>s</sub>), and specific humidity (Q). For all-sky radiance assimilation, additional five hydrometeor analysis variables are introduced: the mixing ratios of cloud water (Qc), cloud ice (Qi), rainwater (Qt), snow (Qs), and graupel (Qg). Following Guerrette et al. (2023) and Jung et al. (2024), linearized hydrostatic balance constraint is used when computing increments of dry-air density (pd) and three-dimensional pressure (P) from the increments of T, Q, and Ps. After the ensemble update, the analysis variables are transformed to MPAS-A's prognostic variables.

Deleted: analysis procedure

Deleted: and applied to assimilate

Deleted: s

Deleted: the

Deleted: it is necessary to

Deleted: first

Deleted: for

Deleted: and credible

Deleted: evaluate

Deleted: DA

Deleted: within the

Deleted: developed by Frolov et al. (2024),

#### 2.1.2 Covariance inflation

There are three multiplicative covariance inflation schemes available within JEDI, including one prior inflation scheme and two posterior inflation schemes with the relaxation to prior perturbation (RTPP; Zhang et al. 2004) and the relaxation to prior spread (RTPS; Whitaker and Hamill, 2012).

The prior inflation scheme multiplies the background error covariance by a factor  $\alpha > 1$ :

$$\mathbf{X}_{inf}^{b} \left( \mathbf{X}_{inf}^{b} \right)^{\mathrm{T}} = \alpha \mathbf{X}_{pri}^{b} \left( \mathbf{X}_{pri}^{b} \right)^{\mathrm{T}}, \tag{1}$$

where **X** is the matrix whose columns are deviation of the ensemble members from the ensemble mean, and the subscript *pri*and *inf* denote prior (i.e., before inflation) and inflated, respectively. The RTPP scheme blends the background and analysis ensemble perturbations as

$$\mathbf{X}_{inf}^{a} = \alpha_{rtpp} \mathbf{X}^{b} + (1 - \alpha_{rtpp}) \mathbf{X}^{a} , \qquad (2)$$

where superscripts a and b denote analysis and background quantities, and  $\alpha_{rtpp}$  denotes the relaxion parameter of RTPP. In the RTPS scheme, the analysis perturbations are inflated as

$$\mathbf{X}_{inf}^{a(i)} = (\frac{\alpha_{rtps}\sigma^{b(i)} + (1 - \alpha_{rtps})\sigma^{a(i)}}{\sigma^{a(i)}})\mathbf{X}^{a(i)}$$
, (3)

where  $\alpha_{rtps}$  is the relaxation parameter of RTPS, the superscript (i) indicates a quantity relevant to the ith element of the state vector  $\mathbf{x}$ , and  $\sigma$  is the ensemble spread. The inflation parameter  $\alpha$  in both RTPP and RTPS is between 0 and 1, where  $\alpha = 0$  indicates no inflation and  $\alpha = 1$  corresponds to relaxing to the background ensemble spread. Within MPAS-JEDI's LGETKF, the three inflation schemes can be employed either individually or in various combinations. Posterior inflation is used in this study.

#### 2.1.3 Covariance localization

The R-localization approach (Hunt et al. 2007) is employed for the observation-space horizontal localization in LGETKF. In this approach, the observation error covariance,  $\mathbf{R}$ , is inflated as a function of the distance between the analysed local volume and surrounding observation locations. The localization is typically performed by multiplying the inverse of a correlation-like function, such as Box Car, Gaspari-Cohn (Gaspari and Cohn, 1999), or the second order auto regressive (SOAR), to the diagonal elements of  $\mathbf{R}$ . Consequently, the influence of observations to the state variables within a local volume decreases smoothly with increasing distance, and observations beyond the specified horizontal localization scale have no influence on the model state update. To speed up the horizontal localization operation, only a limited number of observations (e.g., 1000 used in this study) closest to a local volume is included in the horizontal localization. Additionally, different horizontal localization scales can be applied for different observation types.

The vertical localization in LGETKF is done in model space through the modulated ensemble members it introduces. There are several vertical coordinate options available for vertical localization, including the model level, height, and scale height (log(P)). To reduce the computational cost, it is a common practice, when constructing the modulated ensemble, to approximate the vertical localization matrix using only its first several dominant eigenvectors. In this study, we retain 11 eigenvectors (and thus 11 modulated members for each original ensemble member), which explain ~95% of the variance of our chosen Gaspari-Cohn localization matrix with a 6 km scale.

#### 2.2 LGETKF analysis procedure

As shown in Figure 1, the LGETKF analysis procedure is split into three steps: the observer step for the ensemble background, the solver step for ensemble update, and the observer step for ensemble analysis. In the observer step for the ensemble background (denoted as "OMB"), the model equivalent of the observations (hereafter HofX) is computed for both the original ensemble background members and their corresponding modulated members and then written out into separate files for each ensemble member. The OMB step is carried out with multiple simultaneous jobs, in which the HofX application is looping for a trunk of original and modulated members. The solver step with a single job then reads in the outputs from the OMB step to solve the LGETKF, generating ensemble analysis members. In the observer step for the ensemble analysis (denoted as 150 "OMA"), HofX calculations are performed solely for the original analysis members, omitting the modulated members. Both OMB and OMA steps employ the "Round robin" observation distribution, ensuring an effective load balancing of observations across parallel processing elements (PEs). The "Halo" observation distribution must be utilized in the solver step, storing overlapping sets of observations on each PE and eliminating the need for inter-PE communications by ensuring all required observations for updating model states within a PE available locally. More details on "Round robin" and "Halo" can refer to Frolov et al. (2024).

Note that the OMB and solver steps can be configured to run within a single job execution. In that case, the OMB step must

also use the Halo distribution and the HofX applications for all original and modulated members will have to be run one by one, which leads to a much longer time-to-completion of LGETKF, when compared to the separate job strategy described above. The observation count is another important factor for the memory use and computational speed of both the observer and solver steps. To further improve the computational efficiency, a quality control (QC) step applying HofX to the ensemble mean background is run before the ensemble OMB step. This QC step writes out HofX files (one file for one observation type) with QC flags indicating bad and thinned observations, and then an offline program reads these HofX files and writes out new size-reduced observation by excluding observations that fail QC or are not used after data thinning. The smaller observation files are then used in the subsequent OMB, solver, and OMA steps. Note that the OMA step is mainly for analysis diagnostics, 165 e.g., comparing the analysis fit to observations with the background fit to observations. Therefore, the ensemble forecast step for proceeding to the next cycle can begin before launching the OMA step. Recently, a new function named, "reduce obs space" has been introduced in UFO. This function filters out observations that fail pre-processing checks (e.g., thinning, domain checks) directly in memory during runtime, avoiding unnecessary HofX calculation and storage. Its application is worth

Deleted: bad and thinned

Formatted: Font: Not Bold

exploring as a way to eliminate the separate QC step in the current workflow. In addition, the linear operator is now available for computing ensemble HofXs, following Lei et al. (2018), which is expected to significantly accelerate the Observer step.

Figure. 1. Flow chart of the analysis procedure of LGETKF.

The 80-member MPAS-JEDI's LGETKF experiments configured with a global quasi-uniform grid spacing of 60 km (see Section 3) are conducted on the NCAR's supercomputer Derecho. Table 1 summarizes the wall-clock times of the three job steps of LGETKF at the first analysis time. The QC step, involving HofX of the ensemble-mean background and the processing

of more than 3 million observations, takes about 5.5 minutes when using 4 nodes and 128 cores per node. For each ensemble member, the OMB step consists of HofXs of that background member and its 11 modulated members for the remaining 563197 observations; these calculations need about 5 minutes when using a single node. The solver step updating all 80 members together takes about 25 minutes when using 12 nodes and 64 cores per node. Without these modifications to the analysis procedure, the OMB step takes over 90 minutes to finish the calculation of HofXs for all members and their modulated members, and the Solver step takes about 32 minutes due to the larger observation file read-in. This demonstrates the effectiveness of the improved analysis procedure in MPAS-JEDI's LGETKF. However, the Solver step remains relatively slow and may be not yet feasible for potential operational applications. While using more processors could reduce the run time, the increasing I/O burden for reading observation files limits the achievable performance gains. Active development is ongoing to further improve the computational efficiency of JEDI in general and we anticipate that the time-to-completion of MPAS-JEDI's LGETKF will be further reduced in future versions.

Table 1. Computational resources for different steps of LGETKF at 0000 UTC 15 April 2018.

| Key steps of LGETKF<br>analysis | Nodes/<br>cores per code | Wall-clock Time | Total number of obs. |
|---------------------------------|--------------------------|-----------------|----------------------|
| QC step                         | 4/128                    | 344 seconds     | 3087116              |
| OMB step for one original       |                          |                 |                      |
| and eleven modulated members    | 1/128                    | 286 seconds     | 563197               |
| Solver step                     | 12/64                    | 1470 seconds    | 563197               |

## 3. MPAS-A model configuration, observations, and experimental design

#### 3.1 MPAS-A model configuration

A modified version of MPAS-A version 7.1 is employed as the NWP model in this study, similar to that used by Liu et al. (2022). MPAS-A is a non-hydrostatic model, which is discretized on an unstructured centroidal Voronoi horizontal mesh with C-grid staggering of the state variables for both global and regional applications (Skamarock et al., 2012; 2018). The quasi-uniform global mesh at a grid spacing of ~60 km is used in this study with 163,842 horizontal cells, 55 vertical levels, and a model top height of 30 km. The time step is 360 seconds. The "mesoscale reference" suite is employed for physical parameterizations, utilizing the parameterization schemes detailed in Table 2 in Liu et al. (2022).

Deleted: 286 seconds

Deleted: D

Deleted: expect

Deleted: including the New Tiedtke cumulus scheme (Tiedtke 1989; Zhang and Wang 2017), the WSM6 microphysics scheme (Hong and Lim 2006), the unified Noah land surface model (Chen and Dudhia 2001), the Yonsei University (YSU) planetary boundary layer scheme (Hong et al. 2006), the Monin-Obukhov surface layer scheme (Jimenez et al. 2008), the Rapid Radiative Transfer Model for GCMs (RRTMG) longwave and shortwave radiation schemes (Iacono et al. 2008), the Xu-Randall cloud fraction scheme (Xu et al. 1996), and the YSU orographic gravity-wave drag scheme (Choi and Hong 2015)....

#### 3.2 Observations

summarized in Table 2.

A variety of observations are assimilated in this study, including radiosondes (temperature, zonal and meridional wind components, specific humidity), surface pressure, satellite-derived atmospheric motion vectors (AMV; zonal and meridional wind components), Global Navigation Satellite System Radio Occultation (GNSS RO) bending angle, as well as satellite radiances of selected channels from the Advanced Microsoft Sounding Unit-A (AMSUA-A) and Microwave Humidity Sounder (MHS). Use of GNSS RO bending angle instead of refractivity is a common practice at operational centres (Healy and Thepaut 2006; Rennie 2010; Cucurull et al. 2013) and improves on the assimilation of refractivity that was employed in previous MPAS-JEDI studies (Ivette, personal communication 2024). All experiments assimilate clear-sky AMSU-A radiances from six satellites (NOAA-15/18/19, Aqua, Metop-A/B) and clear-sky MHS radiances from 4 satellites (NOAA-18/19, Metop-A/B). In addition, AMSU-A window-channel radiances from five satellites are assimilated over the water using the all-sky approach in the all-sky radiance DA experiments. Specifically, for clear-sky AMSU-A, radiances from channels 5 (53.596 ± 0.115 GHz), 6 (54.4 GHz), 7 (54.94 GHz), 8 (55.5 GHz), and 9 (57.290344 GHz) are assimilated. For clear-sky MHS, channels 3 (183.311 ± 1.00 GHz), 4 (183.311 ± 3.00 GHz), and 5 (190.311 GHz) are assimilated. For all-sky AMSU-A, channels 1 (23.8 GHz), 3 (31.4 GHz), 4 (52.8 GHz), and 15 (89 ± 1.00 GHz) are assimilated. The assimilated clear-sky and all-sky channels for different sensors and satellites are

Deleted: centers

Deleted: given

Table 2. AMSU-A and MHS channels assimilated using clear-sky or all-sky approach.

| Satellites | Sensors | Clear-sky channels | All-sky channels |
|------------|---------|--------------------|------------------|
| NOAA-15    | AMSU-A  | 5, 7, 8, 9         | 1, 3, 4,15       |
| NOAA-18    | AMSU-A  | 5, 6, 7, 8, 9      | 1, 3, 4,15       |
|            | MHS     | 3, 4, 5            | /                |
| NOAA-19    | AMSU-A  | 5, 6, 7, 9         | 1, 3, 4,15       |
|            | MHS     | 3, 4, 5            | /                |
| Aqua       | AMSU-A  | 8, 9               | /                |
| Metop-A    | AMSU-A  | 5, 6, 9            | 1, 3, 4,15       |
|            | MHS     | 3, 4, 5            | /                |
| Metop-B    | AMSU-A  | 8, 9               | 1, 3, 4          |
|            | MHS     | 3, 4, 5            | /                |

The experiments of Liu et al. (2022), Guerrette et al. (2023), and Jung et al. (2024) assimilated observations that were taken from so-called "GSI-ncdiag" files and had been pre-processed by the GSI DA system, including QC, radiance thinning, and radiance bias correction. Here, we begin from the raw observations (except for MHS radiances) and perform QC, observation error modeling, and bias correction of satellite radiances within MPAS-JEDI using the functions from JEDI's Unified Forward Operators (UFO) configured to mimic the QC procedures in GSI. The MHS satellite radiance observations are still taken from the GSI-ncdiag files, but the bias correction is done within MPAS-JEDI. Observations are converted to IODA from their native format using obs2ioda (https://github.com/NCAR/obs2ioda).

# 3.2.1 Quality control

For all observations, the "PreQC" and "Background Check" filters are applied. The "PreQC" filter discards observations whose "PreQC" quality flags (provided in the observation conversion step for most observations or by GSI for MHS radiance data) are greater than zero for radiance data and 3 for other types of observations. The "Background Check" is employed to reject the observations whose absolute departure from the background exceeds three times the observation error standard deviation. For the surface pressure, a terrain correction scheme (Ingleby 2014) is applied to correct the model-diagnosed surface pressure to the height of the surface station, and an additional QC is applied to reject observations where the differences between surface station height and model terrain height exceed 200 m. GNSS RO bending angle observations are only assimilated below the 30 km model top and use height- and regionally dependent error variances estimated within UFO via the method of Desroziers et al. (2005). Satellite radiances and AMVs are thinned to 145 km spacing to reduce the spatial correlations. Following Guerrette et al. (2023), the observation errors of satellite AMVs are height dependent.

Quality control for clear-sky AMSU-A radiances follows Zhu et al. (2016). First, the cloud liquid water (CLW) content is retrieved from AMSU-A channels 1 and 2, following Grody et al. (2001). Pixels with CLW contents exceeding Care excluded in channels 5 and 6. Next, a thick-cloud filter is applied to remove radiances affected by thick clouds, which are identified through the differences between observations and background in both CLW and radiances. Finally, radiances affected by precipitation are identified and filtered out using the scattering index (Grody et al., 1999). More details of AMSU-A quality control procedures can be found in Zhu et al. (2016).

## 50 3.2.2 Radiance bias correction

The radiance bias b of each channel at different locations is modelled within UFO as a weighed sum of predictors:

$$b = \beta_0 + \sum_{k=1}^K \beta_k p_{k'}, \tag{4}$$

where  $\beta_0$  is the constant offset,  $p_k$  the kth predictor, and  $\beta_k$  the corresponding bias correction coefficient. The bias predictors used in this study are the temperature lapse rate and its square, surface emissivity, and sensor scan angle and its second, third, and fourth powers. In this study, the bias correction coefficients at each LGETKF analysis cycle are obtained from an existing

Deleted: q

Deleted: use

Deleted: a significant

Deleted:

1-month cycling hybrid-3DEnVar experiment, in which the bias correction coefficients are updated across the DA cycles with the variational bias correction (VarBC; Dee, 2004) technique.

#### 3.2.3 All-sky radiance assimilation

The LGETKF of MPAS-JEDI has the capability of assimilating radiances using the all-sky approach, following the same procedures described by Liu et al. 2022 for MPAS-JEDI's variational DA methods. Three key ingredients for all-sky radiance DA include the introduction of hydrometeors as part of analysis variables, the cloudy radiance observation operator via 275 Community Radiative Transfer Model (CRTM; Benjamin et al. 2023), and situation-dependent all-sky observation error models (Geer and Bauer 2011). Since the WSM6 microphysics scheme is employed, the mixing ratios of Qc, Qi, Qr, Qs, and Qg are used as the hydrometeor analysis variables. For the window channels assimilated using the all-sky approach, only overwater radiances are assimilated due to the large uncertainty of surface emissivity over the land, snow, and ice surfaces. As in Liu et al. (2022), the all-sky observation error for each AMSU-A window channel is modelled as a piecewise linear ramp function of the average of cloud liquid water (CLW) path retrieved from the observed and CRTM-simulated (from the background) radiances of AMSU-A channels 1 and 2. For each channel, two CLW thresholds are defined. Observations with an averaged CLW below the lower threshold are treated as clear-sky and assigned the clear-sky observation error, while those above the upper threshold use the all-sky observation error. For averaged CLW values between the thresholds, the observation error increases linearly from the clear-sky to the all-sky observation error as the averaged CLW increases. This approach makes the all-sky observation error cloud-dependent, accounting for cloud-related uncertainties arising from both the observations and the background. More details about the all-sky observation error model are given by Liu et al. (2022).

#### 3.3 Experimental design

Six 6-hourly cycled 80-member ensemble analysis and forecasting experiments are conducted as summarized in Table 3.

Experiment AllSky serves to evaluate the impact of all-sky radiance DA compared to ClrSky, the clear-sky radiance DA experiment. The other four experiments explore different settings for covariance inflation and localization, and motivate the specific combination of the inflation and localization used in ClrSky and AllSky.

All experiments begin from a time-lagged ensemble background valid at 0000 UTC 15 on April 201 The time-lagged ensemble background is generated using four sets of 20-member ensemble forecasts of different lead times (24-h, 18-h, 12-h, and 6-h), initialized from Global Ensemble Forecast System (GEFS) ensemble analyses valid at 0000, 0600, 1200, and 1800 UTC on 14 April 2018, respectively. The assimilation period for the experiments ClrSky and AllSky is nearly one month, with the last analysis cycle being 1200 UTC on 14 May 2018, while the other experiments span the first 10 days of that period.

In ClrSky, the non-radiance observations and clear-sky radiances from AMSU-A temperature sounder channels and MHS water vapor channels are assimilated (see Table 2). For the rest of the experiments, radiances from AMSU-A window channels under all-weather situations are assimilated in addition to those assimilated in ClrSky.

Deleted: n
Deleted: initial
Deleted: 8,
Deleted: which

The AllSky-RTPS, AllSky-RTPP, and AllSky are conducted to evaluate the sensitivity to the covariance inflation with the same setting for the covariance localization. Following Guerrette et al. (2023), the RTPS is employed in AllSky-RTPS with  $\alpha$ \_RTPS=1.0, and RTPP is used in AllSky-RTPP with  $\alpha$ \_RTPP = 0.7. Following Met Office's EDA system (Bowler et al. 2017), RTPP and RTPS are applied sequentially in AllSky, with  $\alpha$ \_RTPP=0.5 and  $\alpha$ \_RTPS=0.9. The AllSky, AllSky-L1200, and AllSky-L600 use the same setting for the inflation but use horizontal localization scales of 300 km, 1200 km, and 600 km, respectively, for all-sky AMSU-A channels. In all experiments, the horizontal localization scale is set to 1200 km for the non-radiance data and clear-sky AMSU-A and MHS radiances, following Liu et al. (2022) and Guerrette et al. (2023), while the vertical localization scale is set to 6 km.

Table 3. Inflation and horizontal localization configurations for six experiments

| Experiments  | Inflation                                     | Horizontal localization scale (km) |                 |
|--------------|-----------------------------------------------|------------------------------------|-----------------|
|              |                                               | Clear-sky channel                  | All-sky channel |
| ClrSky       | $\alpha_{RTPS} = 0.9,  \alpha_{RTPP} = 0.5$   | 1200                               | /               |
| AllSky       | $\alpha_{RTPS}=0.9,\alpha_{RTPP}=0.5$         | 1200                               | 300             |
| AllSky-RTPS  | $\alpha_{RTPS} = 1.0$                         | 1200                               | 300             |
| AllSky-RTPP  | $\alpha_{RTPP} = 0.7$                         | 1200                               | 300             |
| AllSky-L1200 | $\alpha_{RTPS}=0.9,\alpha_{RTPP}=0.5$         | 1200                               | 1200            |
| AllSky-L600  | $\alpha_{RTPS} = 0.9$ , $\alpha_{RTPP} = 0.5$ | 1200                               | 600             |

#### 4 Results from inflation and localization sensitivity experiments

Due to the limited ensemble size, the EnKF-based methods suffer from sampling errors, which can lead to filter divergence and degraded analysis. To combat the sampling errors, the covariance inflation and localization techniques are commonly employed in the EnKF-based methods. In this section, we provide the results for the sensitivity to the covariance inflation setting first and then for the sensitivity to the horizontal localization setting for all-sky AMSU-A radiances.

## 4.1 Sensitivity to covariance inflation configuration

Many studies have emphasized the sensitivity of analysis performance to the choice of inflation parameters (e.g., Whitaker and Hamil, 2012; Nerger, 2015). Kotsuki et al. (2017) found that using either RTPP or RTPS alone could lead to an over-dispersive ensemble in certain regions, and that this behavior could be resolved by using a combination of additive and multiplicative inflations. Bowler et al. (2017) found that a combination of RTPS and RTPP could maintain ensemble spread and reduce forecast errors in the Met Office's EDA system. Given that the additive inflation is not implemented yet in the

current JEDI system, we only examine the multiplicative inflation techniques with RTPP or RTPS alone in AllSky-RTPP and AllSky-RTPS, or with a combination of RTPP and RTPS in AllSky.

Figure 2 shows the time series of the model-space ensemble spread and ensemble-mean root-mean-square error (RMSE) of the background 6-h forecast for the vertically-averaged U, V, T, and water vapor mixing ratio (Qv) fields of the three experiments, with the GFS analyses treated as the truth. Due to the temperature bias near the model top (Liu et al. 2022), the evaluation of T is limited to below model level 50 (~25 km AGL). Both AllSky-RTPP and AllSky-RTPS, applying RTPP or RTPS alone, present a consistent trend that the ensemble spread (solid black and blue curves) gradually decreases with assimilation cycles for all the four variables. Consequently, the ensemble-mean RMSE gradually increases over time. In contrast, the ensemble spread in AllSky with a combination of RTPP and RTPS is the largest and well maintained over time, with the corresponding ensemble-mean RMSE being the smallest among the three experiments. This result is consistent with Bowler et al. (2017) using an EDA system.

Deleted:

- Figure 2. Time series of the model-space ensemble spread (solid lines) and ensemble-mean RMSE (dashed lines) of the background 6-h forecast verified against the GFS analyses for the vertically-averaged (a) U, (b) V, (c) T, and (d) Qv fields from the three experiments. Statistics are calculated globally below model level 50 between 0000 UTC 18 and 0000 UTC 25 April 2018 at an interval of 6 hours.
- The benefits of employing a combination of RTPP and RTPS are also evident when verified against observations. Figure 3 shows the vertical profiles of the observation-space total spread (i.e., square root of the sum of the ensemble variance and observation error variance) and ensemble-mean RMSE of the background, verified against radiosonde observations. For a good ensemble DA system, the total spread is expected to be comparable with the ensemble-mean RMSE. AllSky outperforms AllSky-RTPP and AllSky-RTPS by producing the largest total spread and the smallest ensemble-mean RMSEs in all observed variables at almost all levels. However, the total spread smaller than the ensemble-mean RMSE for certain levels of U, V, and T also suggests that further improvement is needed by tuning observation errors and/or inflation/localization settings in a future study. This issue has also been observed in MPAS-JEDI's EDA system (Guerrette et al. 2023).
  - schemes (Buizza et al. 1999; Berner et al. 2009) in the current MPAS-A model. It is anticipated that the ensemble DA performance with MPAS-JEDI could be improved by introducing stochastics physics or using a multi-physics approach in the ensemble forecast step. In addition, optimal inflation configurations may depend on factors such as ensemble size, mesh resolution, and other DA settings. Developing robust and generally applicable inflation strategies therefore requires further systematic investigation.

It is worth mentioning that the model errors are not taken into account due to the lack of stochastics physics parameterization

Deleted:

Figure 3. Vertical profiles of the observation-space ensemble total spread (solid lines) and ensemble-mean RMSE (dashed lines) of the background verified against radiosonde (a) U, (b) V, (c) T, and (d) Q observations. Statistics are calculated globally from the ensemble background between 0000 UTC 18 and 0000 UTC 25 April 2018 at an interval of 12 hours.

## 4.2 Sensitivity to horizontal localization scale for all-sky radiances

In the global EnKF-based systems, it is a common practice to use a relatively large horizontal localization scale for satellite radiances (e.g., 1200 km in Lei et al., 2020 and 1000 km in Bonavita et al., 2020). Previous studies also show the benefits of using a shorter horizontal localization scale for all-sky satellite radiances in the regional EnKF-based systems. Okamato et al. (2018) found that using a horizontal localization scale of 365 km for Himawari-8 radiance assimilation improved the fit of the

370 background to radiosonde observations compared to using a horizontal localization scale of 730 km in their regional LETKF system. Bonavita et al. (2020) also showed that reducing the horizontal localization scale for all-sky satellite radiances from 2000 km to 1000 km produced ~1% improvement in ECMWF's global EnKF system. Therefore, we further evaluate the sensitivity of horizontal localization scales for AMSU-A window channel all-sky radiances by comparing AllSky-L1200, AllSky-L600, and AllSky.

Figure 4 illustrates the percentage difference in the ensemble-mean RMSE of U, V, T, and Qv, verified against the GFS analyses, for AllSky-L600 and AllSky relative to AllSky-L1200, at different model level and latitude bins. Following Guerrette et al. (2023), the bootstrap resampling method is employed to calculate the confidence intervals of the differences. The samples are resampled 10000 times with replacement, and the 95 % confidence intervals are obtained using the percentile method (Gilleland et al., 2018), corresponding to the 2.5th and 97.5th percentiles of the bootstrap distribution. It can be clearly seen that using a 600 km localization scale in AllSky-L600 reduced RMSEs for all the four variables at most latitude bins and model

that using a 600 km localization scale in AllSky-L600 reduced RMSEs for all the four variables at most latitude bins and model levels, with a larger improvement over the southern hemisphere, when compared to AllSky-L1200 with a 1200 km localization scale. Decreasing the localization scale to 300 km in AllSky further improved the short-term forecasts as indicated by model/latitude bins with darker blue colour in the right column of Figure 4. Given the overall superior performance of using 300 km localization scale, and considering that 300 km is sufficiently small relative to the model mesh resolution of 60 km, we did not further test shorter horizontal localization scales, even though they might offer further improvements. Consequently,

the AllSky experiment was extended to cover the full month and is compared with the ClrSky experiment in the next section.

Deleted: than

Deleted: is

Deleted: for

Figure 4. Left: RMSE of AllSky-L1200's ensemble mean background for U, V, T, and Qv; Middle/Right: percent difference in RMSE for AllSky-L600 and AllSky, relative to AllSky-L1200. The RMSEs are computed using the GFS analyses as the reference and the statistics are binned with 5 model levels and 11° latitude bands. Inset black circles in individual bins indicate that the difference in RMSE between experiments is statistically significant with a 95% confidence interval, as determined via bootstrap resampling from 29 samples spanning 0000 UTC 18 April to 0000 UTC 25 April 2018 at 6-hour intervals.

**Deleted:** for the period from 0000 UTC 18 April to 0000 UTC 25 April 2018...

## 5. Impact of all-sky radiance DA

This section evaluates the impact of all-sky AMSU-A radiance DA for the short-term background forecasts (i.e., 6-h lead time) first and then for the extended-range forecasts up to 7 days.

## 5.1 6-hour forecasts

Figure 5 shows the percent RMSE difference of the ensemble mean of 6-h forecasts for U, V, T, and Q<sub>v</sub> of AllSky relative to ClrSky, when verified against the GFS analyses. Positive impacts (model level/latitude bins with blue colour) occur for U, V, and Q<sub>v</sub> almost everywhere between 50°S and 50°N, with largest improvements in the tropics. There are some degradations for these three variables at the high latitudes and below the model level 30 (for U and V) or 20 (for Q<sub>v</sub>) over the southern hemisphere. Largest degradation is seen in T in the latitude band of 50°S-60°S and between model levels 5 and 15.

Figure 5. Percent difference in RMSE for AllSky relative to ClrSky for (a) U, (b) V, (c) T, and (c) Qv of the ensemble mean of 6-h forecasts, Inset black circles in individual bins indicate that the difference in RMSE between experiments is significant with a 95% confidence interval, as determined via bootstrap resampling from 104 samples spanning 0000 UTC 18 April to 1800 UTC 14 May 2018 at 6-hour intervals.

Deleted: are

**Deleted:** from 0000 UTC 18 April to 1800 UTC 14 May 2018 at an interval of 6 hours

Additional evidence of temperature degradation associated with all-sky radiance assimilation is observed when verified against clear-sky radiances. Figure 6 shows the latitude-binned RMS and mean of ensemble mean background minus observations for bias-corrected clear-sky radiances from five AMSU-A temperature sounding channels aboard NOAA 18. Assimilation of AMSU-A window channels produces more noticeable differences between ClrSky and AllSky in channels 5, 6, and 7, which are sensitive to lower and middle atmospheric levels where clouds and precipitation are prevalent. As expected, AllSky reduces the mean value statistics in these channels because the inclusion of more cloud-affected observations reduces the simulated 425 brightness temperatures. Notably, at lower latitudes, AllSky yields slightly smaller RMS values and mean OMB values closer to zero, indicating benefits of all-sky assimilation. However, between 50°S and 70°S, AllSky exhibits a more pronounced cold bias than ClrSky in channel 5, accompanied by a marked RMS increase, consistent with the temperature degradation highlighted in Figure 5. In contrast, channels 8 and 9, which are less sensitive to cloud-affected layers, show only minor differences between both experiments,

Figure 6. RMS (solid line) and mean (dashed line) of the ensemble mean background minus observations, binned by latitude, for bias-corrected clear-sky radiances from NOAA 18,AMSU-A: (a) channel 5, (b) channel 6, (c) channel 7, (d) channel 8, and (e) channel

Deleted: observations minus Deleted: (OMB) Deleted: statistics Deleted: 0 Deleted: shows larger Deleted: OMB Deleted: due to Deleted: increase in Deleted: and it is

Deleted: observations minus

Deleted: (i.e., OMB)

Deleted: 0

Deleted:

Deleted: 5

9, for CIrSky (blue) and AllSky (red). Statistics are calculated globally every 6 hours from 0000 UTC 18 April to 1200 UTC 14 May 2018, using  $10^{\circ}$  latitude bins.

The temperature degradation in the southern midlatitudes aligns with challenges documented in other systems. A similar pattern was reported in the ECMWF's 4DVar system when assimilating all-sky AMSU-A channel 4 radiances (Weston et al. 2019). Lonitz and Geer (2015) attributed such issue to model deficiencies in representing liquid water in cold-sector clouds over the Southern oceans—a microphysical bias likely presents in other forecast systems as well (Tong et al., 2020). Given these similarities, we suspect that MPAS also suffers from similar microphysical bias. To address this, a screening procedure for such cold-sector clouds, following Lonitz and Geer (2015), could be developed for MPAS-JEDI. Note that the current bias correction scheme for all-sky radiances is still based upon the existing clear-sky approach and updated by an independent cycling deterministic DA system, which is sub-optimal. We plan to introduce cloud-/precipitation-related predictors for the bias correction of all-sky radiance DA, which has been shown to improve all-sky infrared radiance DA (Okamoto et al., 2023). Implementation of an online bias correction method, where correction coefficients are included as part of the analysis variables (Fertig et al., 2009; Miyoshi et al., 2010; Chandramouli et al., 2022), is also under consideration for MPAS-JEDI's LGETKF.

### 5.2 7-day forecasts

We initialized 7-day forecasts from the ensemble mean analysis at 0000 UTC each day from 18 April to 14 May 2018, for a total of 27 forecasts. The first three days (15-17 April) are treated as a spin-up period for the DA cycles to reach MPAS-JEDI's own cycling climatology, and are excluded from the extended forecast evaluation.

## 5.2.1 Model space verification

The relative RMSE differences of AllSky from ClrSky as a function of forecast lead time, when verified against GFS analyses for the vertically averaged upper-air fields over different regions, are shown in Figure 7. The largest percent improvement from AllSky occurs for Qv with a 3-6% RMSE reduction for the day-1 forecast over the tropical regions. The statistically significant improvement for Qv lasts up to 7 days over the ITCZ and 5-6 days over northern and southern tropical regions. A smaller positive impact of AllSky on Qv lasts up to 3 days over the southern extratropic region with a mostly neutral impact over the northern extratropic. Similar positive impact of AllSky can also be seen for the U- and V-wind components with the statistically significant improvement lasting up to 7 days for the U-wind component, but only up to 5 days for the V-wind component over the ITCZ and northern tropical regions. Consistent with some short-term forecast degradation in T (Figure 5c) in AllSky, the negative impact is also seen for the first 2-3 days, and then the impact becomes neutral and even positive over some regions, although not statistically significant.

Deleted: ,

Deleted:

Figure 7. The differences in RMSE of the upper-air (a) U, (b) V, (c) T, and (d) Qv between ClrSky and AllSky as a function of forecast lead time and regions, using GFS analyses as reference. Regions are defined as NX (northern extratropic; 30°N-90°N), NT (northern tropical; 5°N-30°N), ITCZ (intertropical convergence zone; 5°S-5°N), ST (southern tropical; 5°S-30°S), and SX (southern extratropic; 30°S-90°S). The upward (downward) black triangles indicate the statistically significant improvements (degradations) at the 95% confidence level, as determined via bootstrap resampling from 27 daily samples spanning 18 April to 14 May 2018.

Figure 8 shows the relative RMSE differences between ClrSky and AllSky for surface variables verified against GFS analyses. Consistent with the impact on the upper air variables in Figure 7, the largest and most significant positive impact from AllSky is on 2-meter  $Q_v$  with a ~2.8% RMSE reduction for the day-1 forecasts and a ~0.5% RMSE reduction still remaining for the day-6 forecasts. The positive impact of AllSky on the 10-meter U-wind component is also evident with a smaller variation of the percent RMSE reduction (~0.5% - ~1%) for the first five days, but less significant than the impact on  $Q_v$ . Similar to the upper-air T forecasts, the all-sky DA also slightly degrades the 2-meter T forecasts for the first 4 days and then improves forecasts from day-5. For the surface pressure forecasts, the AllSky experiment mostly outperforms the ClrSky experiment for all the forecast ranges, except for the first day forecast, with a maximum improvement of 2% at day-5.

Figure 8. Similar to Figure 7, but for the global statistics of percent RMSE differences for (a) 10-meter U (U10), (b) 2-meter T (T2), (c) 2-meter Q<sub>v</sub> (Q2), and (d) surface pressure (Ps). The bars below or above zero indicate that the difference is statistically significant at a 95% confidence interval, as determined via bootstrap resampling from 27 daily samples spanning 18 April to 14 May 2018.

These model-space verification results for AllSky vs. ClrSky are overall consistent with those of Liu et al. (2022), who evaluated the impact of all-sky AMSU-A radiance DA above the clear-sky AMSU-A radiances using MPAS-JEDI's 3DEnVar. This study with MPAS-JEDI's LGETKF evaluates the impact of all-sky AMSU-A radiance DA above clear-sky radiances from both AMSU-A and MHS, i.e., a more challenging impact assessment with more clear-sky radiance data in the benchmark experiment, leading to a smaller magnitude of improvement than that of Liu et al. (2022).

# 5.2.2 Observation space verification

The forecast impacts of all-sky DA are further evaluated with radiosonde observations and independent (i.e., not assimilated)
Advanced Technology Microwave Sounder (ATMS) radiances in all-weather situations. Similar to the upper-air model-space
verification, all-sky AMSU-A radiance DA improves the forecasts of U, V, and Q, but degrades the forecasts of T, when
verified against the radiosonde observations (Figure 9). While all-sky AMSU-A radiances are assimilated only over water,

radiosonde data are mostly available over land. It could take several days for the positive impacts to propagate from the seas to the land areas, likely leading to the positive impact being the largest in the day-5 or day-6 for the forecasts of U, V, and Q, compared to the largest positive impacts observed in the first day in the global average in Figure 7.

 $Figure~9.~Similar~to~Figure~8, but~verified~the~forecasts~against~radios onde~(a)~U,\\ (b)~V,\\ (c)~T,~and~(d)~Q~observations.$ 

Figure 10 shows the relative forecast RMSE differences between AllSky and ClrSky, when verified against brightness temperatures of nine ATMS channels from NOAA-20. The nine ATMS channels include three window channels (1, 2, and 3) sensitive to cloud and precipitation, three temperature sounding channels (5, 7, and 9), and three water vapor channels (18, 19, and 20). The MPAS-JEDI's HofX application is employed to calculate the simulated brightness temperatures using the cloudy CRTM operator with the day-1 to day-7 forecasts as input. The verification statistics are calculated only over water for the three window channels and over both water and land for the other six channels. The benefit of assimilating all-sky AMSU-A radiances is larger on clouds than on moisture as evidenced with a larger RMSE reduction for the three window channels than for the three water vapor channels. Interestingly, the positive impact of AllSky is found for the three temperature channels throughout the 7-day forecasts, unlike the negative T impact in model space verification for the first 3-day forecasts in Figure 7c. This discrepancy should be caused by the model-space T verification being vertically and globally averaged and the three

ATMS temperature channels sensing different layers of the atmosphere, with the channel 5 for a layer around 850 hPa, the channel 7 around 400 hPa, and the channel 9 around 200 hPa.

Figure 10. Similar to Figure 9, but verified the forecasts against brightness temperatures of nine ATMS channels from NOAA-20.

## 530 6. Conclusions

In this study, the LGETKF algorithm within the JEDI framework is implemented in MPAS-JEDI along with the capability of assimilating clear-sky and all-sky satellite radiance data. As the first implementation and evaluation of MPAS-JEDI's LGETKF, this study investigates its computational behaviour, optimal configuration, and performance through a series of sensitivity experiments and 1-month cycled DA experiments, demonstrating the stable and robust performance of MPAS-

535 JEDI's LGETKF for community research and potential operational applications.

The LGETKF performs vertical localization in model space, making it suitable for assimilating nonlocal observations such as satellite radiances, but with additional cost for the computation of HofXs related to the modulated ensemble members. Performing the Opserver and Opserver and Opserver and Opserver with different jobs allows the use of different parallelization strategies for the

| Deleted: 0 |  |  |
|------------|--|--|
| Dolotodi   |  |  |

observation distribution across processing elements in two steps, which improves the computational efficiency of MPAS-JEDI's LGETKF. Additionally, computing the HofX of each ensemble member in parallel further reduces the overall time required to complete the LGETKF analysis.

The impacts of assimilating over-water all-sky radiances from AMSU-A window channels are investigated with a global 80545 member LGETKF at a quasi-uniform grid spacing of 60 km over a one-month period. The efforts are made to tune the
covariance inflation and horizontal localization first. Results indicate that combining RTPS and RTPP for the inflation is more
effective than using RTPS or RTPP alone in maintaining the ensemble spread over time and performs the best with the smallest
RMSE of the ensemble mean background. The sensitivity experiments also demonstrate the benefits of using a smaller
horizontal localization scale for all-sky AMSU-A radiances. The model-space verification shows the overall positive impacts
of all-sky AMSU-A radiance DA, when compared to the benchmark experiment assimilating clear-sky AMSU-A and MHS
radiances, on the forecasts of moisture and wind fields with the largest improvement over the tropical regions up to 7 days.
All-sky AMSU-A DA also leads to an improved forecast fitting to the ATMS window channels' radiances, indicating a better
forecast of clouds and precipitation. All-sky AMSU-A DA impact from this study is mostly consistent with Liu et al. (2022)
using MPAS-JEDI's 3DEnVar.

One issue from all-sky AMSU-A DA, which was also seen from previous studies, is the degradations of temperature forecast, especially at the lower troposphere around 60°S where the forecasts are likely biased due to the model deficiency in simulating cold-air outbreaks. This issue could be resolved by applying more strict quality control, improving physical parameterization of the model, enhancing radiance bias correction scheme, or revisiting the all-sky observation error model. Despite the efforts for tuning the inflation and the localization configurations, the ensemble of LGETKF is still sub-optimal with insufficient ensemble spreads, especially in the upper troposphere and stratosphere. Further improvements could be made with spatially and temporally adaptive inflation factors (Anderson 2009) or scale-dependent localization (Wang et al. 2022). A better specification of the observation errors using the statistical method of Desroziers et al. (2005) will also improve the analysis accuracy in general. Also, an ensemble-based forecast sensitivity to observations (EFSO; Liu and Kalnay 2008; Kalnay et al. 2012) diagnosis tool has been developed for MPAS-JEDI's LGETKF to estimate the impacts of assimilated observations on short-term forecasts. This tool could be used to further investigate the effects of all-sky assimilation and to enhance the performance of MPAS-JEDI's LGETKF.

MPAS-JEDI's LGETKF has the capability to assimilate all-sky radiance data from a wide range of microwave and infrared sensors, and we plan to explore them in future studies. The regional DA capability of MPAS-JEDI is under active development, and the application of the EnKF-based methods to convective-scale DA is already underway.

Author contributions: TS designed, conducted, and analysed all experiments and wrote the manuscript. JJG initialized the LGETKF implementation in MPAS-JEDI. JB configured the all-sky assimilation of AMSU-A radiances. ZL and CS aided with experimental design and analysis. All co-authors contributed to the development of the MPAS-JEDI, preparation of externally sourced data, design of experiments, and preparation of the manuscript.

Deleted: also

Code and data availability: The source code of MPAS-JEDI 2.1.0 is available on Zenodo at https://doi.org/10.5281/zenodo.15201032 (last access: 29 April 2025; JCSDA and NCAR 2023). Global Forecast System analyses are available from NCAR Research Data Archive (RDA) https://rda.ucar.edu/datasets/ds084.1/ (last access: 21 January 2025; National Centers For Environmental Prediction/National Weather Service/NOAA/U.S. Department Of 580 Commerce, 2015). Global Ensemble Forecast System ensemble analyses downloaded from https://www.ncei.noaa.gov/products/weather-climate-models/global-ensemble-forecast (last access: 21 October 2025). Conventional satellite NCAR and observations assimilated downloaded are from RDA https://rda.ucar.edu/datasets/d337000 (last access: 21 January 2025; National Centers For Environmental Weather Service/NOAA/U.S. Of 585 Prediction/National Department Commerce, 2008) and https://rda.ucar.edu/datasets/d735000/ (last access: 21 January 2025; National Centers For Environmental Prediction/National Weather Service/NOAA/U.S. Department Of Commerce, 2009). ATMS radiance data used in the forecast verification are downloaded from https://sounder.gesdisc.eosdis.nasa.gov/opendap (last access: 21 January 2025).

690 Acknowledgements: The National Center for Atmospheric Research (NCAR) is sponsored by the National Science Foundation of the United States. The numerical calculations of this study are supported by NCAR's Computational and Information Systems Laboratory (CISL). We thank Sergey Frolov, Wei Huang, and Bo Huang in NOAA Physical Sciences Laboratory for valuable discussions.

Competing interests. The authors have no competing interests.

Financial support: This research has been supported by the United States Air Force (grant no. NA21OAR4310383).

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
