# Peer review of "All-sky AMSU-A radiance data assimilation using the gain-form of Local Ensemble Transform Kalman filter within MPAS-JEDI-2.1.0: implementation, tuning, and evaluation"

_EGUsphere, 2025_

## Referee Comment (RC1)

**Review for EGUsphere-2025-2079**

**All-sky AMSU-A radiance data assimilation using the gain-form of the Local Ensemble Transform Kalman filter within MPAS-JEDI-2.1.0: implementation, tuning, and evaluation**

**Specific comments:**

Section 2.2: The offline tool you describe to remove bad or thinned observations based on the ensemble mean HofX sounds similar to what the new "reduce obs space" filter action does in JEDI. Perhaps you should reference this new action here so that other JEDI users are aware of this updated functionality.

Section 2.2: Could you provide some comparisons of the runtime before and after implementing these optimizations to parallelize the HofX calculation. I think a main novelty of this study is the use of MPAS-JEDI, and it would be nice to focus a little more on the improvements you made to that system.

L157: The 25 minute runtime for the solver step is very long and not feasible for operational settings. Does the runtime scale well with increasing nodes or cores? Also, could you add some details on ideas or current progress on how this will be sped up?

L211: "Pixels with a significant CLW content are excluded". This is vague - could you give the exact threshold for CLW?

Fig. 3: It is a little hard to deduce the different lines here when they overlap. Perhaps a plot of consistency ratio would be better? If you choose to keep the same format, I recommend changing the x-axis label to "RMSE, Total Spread" instead of "RMSE/Total Spread".

Section 4.2: Given that the 300 km localization scale produced the best results, do you think lowering it more would further improve results? This doesn't necessitate a new experiment, but it would be nice to comment on the possibility.

L356: It is confusing to state that there is a "pronounced cold bias" but also see that the dashed lines in Fig. 6a are greater than zero below 50 degrees south. I recommend plotting -1*OMB to show the more commonly interpreted version of Bias. I also recommend adding a thicker vertical line for 0 K so that we can better deduce between warm and cold biases. Also, the phrasing here that "AllSky exhibits a pronounced cold (warm?) bias" suggests that ClrSky does not have the same bias. But really it is AllSky that is further increasing the warm bias.

**Technical comments:**

L86: "linearized hydrostatic balance constraint is when" -> "the linearized hydrostatic balance constraint is used when"
L102: Relaxation is misspelled

L156: Saying "about 286 seconds" feels overly precise for something meant to be approximate. To stay consistent with how the other runtimes are described, maybe just say "about 5 minutes" instead.

L169: The list of physical parameterization schemes is a bit long and wordy. I recommend moving this list into a table.

L195: "configured mimic" -> "configured to mimic"

L199: "Quality" should be capitalized in the section name

L240: The sentence beginning "All experiments begin from…" is long and should be split in two.

L313: "observations than using" -> "observations compared to using"

L351: "NOOA" -> "NOAA"

L369: "presents" -> "present"

L376: "LGETK" -> "LGETKF"

L381: "climatology, and excluded" -> "climatology and are excluded"

L384: "vertically averaged" -> "vertically-averaged"

---

## Author Response (AR1)

**Reply to RC1:**

This paper evaluates the impacts of assimilating all-sky radiance observations using the LGETKF implementation within the MPAS-JEDI framework. The LGETKF solver is particularly suitable for this observation type, as it performs model-space localization that does not require an explicit vertical coordinate for the observations. The authors discuss improvements to the computational efficiency of the LGETKF and explore tuning strategies for covariance inflation and localization. Following these developments, the assimilation of all-sky radiance observations in a global MPAS simulation yields improvements in many atmospheric fields, with the exception of temperature.

Overall, the manuscript is clearly written and well structured. While much of the scientific content aligns closely with findings from earlier studies (and therefore may not be especially novel), the paper's main contribution lies in its application of the new JEDI system, particularly the global implementation of LGETKF within MPAS-JEDI. Given the emerging importance of JEDI for both operational and research-oriented data assimilation systems, this study provides timely and valuable insight into the system's performance, optimal configuration, and computational behavior. In this context, I find the manuscript suitable for publication, provided that a few minor issues are addressed (see attached PDF). A slightly stronger focus on the novelty and implications of using the JEDI system would also enhance the paper's contribution.

Reply: We appreciate the reviewer's thoughtful feedback and constructive suggestions. We have carefully revised the Introduction and Conclusion sections to highlight the novelty and contribution of this study.

**Specific comments:**

Section 2.2: The offline tool you describe to remove bad or thinned observations based on the ensemble mean HofX sounds similar to what the new "reduce obs space" filter action does in JEDI. Perhaps you should reference this new action here so that other JEDI users are aware of this updated functionality.

Reply: Thanks for pointing it out. The "reduce obs space" function has been implemented in MPAS-JEDI 3.0.0. We have added additional statements of the function in Lines 151-156 in the revision.

Section 2.2: Could you provide some comparisons of the runtime before and after implementing these optimizations to parallelize the HofX calculation. I think a main novelty of this study is the use of MPAS-JEDI, and it would be nice to focus a little more on the improvements you made to that system.

Reply: Thanks for your suggestion. Without these modifications to the analysis procedure, the OMB step takes over 90 minutes to finish the calculation of HofXs for all members and their modulated members, and the Solver step takes about 32 minutes due to the larger observation file read-in. Related statements have been added in Lines 166-169 in the revision.

L157: The 25 minute runtime for the solver step is very long and not feasible for operational settings. Does the runtime scale well with increasing nodes or cores? Also, could you add some details on ideas or current progress on how this will be sped up?

Reply: We agree that current the Solver step is a little bit slow for potential operational application. This could be improved with more nodes. However, when the number of cores increase, the IO of observations brings additional burden. Related statements have been added in Lines 169-171 in the revision.

L211: "Pixels with a significant CLW content are excluded". This is vague - could you give the exact threshold for CLW?

Reply: The value 0 is actually applied.

Fig. 3: It is a little hard to deduce the different lines here when they overlap. Perhaps a plot of consistency ratio would be better? If you choose to keep the same format, I recommend changing the x-axis label to "RMSE, Total Spread" instead of "RMSE/Total Spread".

Reply: Thanks for your suggestions. We tried to show consistency ratio instead of both RMSE and total spread. However, it doesn't show the gap between RMSE and total spread, which can give more information of the performance of LGETKF. Therefore, we prefer to use RMSE and total spread instead of consistency ratio, with "RMSE, Total Spread" instead of "RMSE/Total Spread" showing in x-axis. Similar modification is done to Figure 2.

Section 4.2: Given that the 300 km localization scale produced the best results, do you think lowering it more would further improve results? This doesn't necessitate a new experiment, but it would be nice to comment on the possibility.

Reply: A shorter localization scale might yield further improvements. However, given the model mesh resolution of 60 km, the 300 km horizontal localization scale is already sufficiently small, so we did not test shorter scales. We have added related statements to Lines 350–351 in the revision.

L356: It is confusing to state that there is a "pronounced cold bias" but also see that the dashed lines in Fig. 6a are greater than zero below 50 degrees south. I recommend plotting -1\*OMB to show the more commonly interpreted version of Bias. I also recommend adding a thicker vertical line for 0 K so that we can better deduce between warm and cold biases. Also, the phrasing here that "AllSky exhibits a pronounced cold (warm?) bias" suggests that ClrSky does not have the same bias. But really it is AllSky that is further increasing the warm bias.

Reply: Thank you for your suggestions. We have modified Figure 6 to use ensemble mean background minus observations instead of OMB to better align with the statements. A reference line at 0 K is added to help distinguish between warm and cold biases. We have also revised the text accordingly, changing "AllSky exhibits a pronounced cold bias" to "AllSky exhibits a more pronounced cold bias."

**Technical comments:**

L86: "linearized hydrostatic balance constraint is when" -> "the linearized hydrostatic balance constraint is used when"

Reply: Fixed.

L102: Relaxation is misspelled

Reply: Fixed.

L156: Saying "about 286 seconds" feels overly precise for something meant to be approximate. To stay consistent with how the other runtimes are described, maybe just say "about 5 minutes" instead.

Reply: Fixed.

L169: The list of physical parameterization schemes is a bit long and wordy. I recommend moving this list into a table.

Reply: Since the parameterization schemes are exactly the same as those in Liu et al. (2022), we prefer to cite Liu et al. (2022) without showing the details.

L195: "configured mimic" -> "configured to mimic"

Reply: Fixed.

L199: "Quality" should be capitalized in the section name

Reply: Fixed.

L240: The sentence beginning "All experiments begin from..." is long and should be split in two.

Reply: We have revised the sentence in Lines 264-265.

L313: "observations than using" -> "observations compared to using"

Reply: Fixed.

L351: "NOOA" -> "NOAA"

Reply: Fixed.

L369: "presents" -> "present"

Reply: Fixed.

L376: "LGETK" -> "LGETKF"

Reply: Fixed.

L381: "climatology, and excluded" -> "climatology and are excluded"

Reply: Fixed.

L384: "vertically averaged" -> "vertically-averaged"

Reply: Fixed throughout the paper.

**Reply to RC2:**

This study used the LGETKF implementation within the MPAS-JEDI system with a global MPAS model and compared the impact of assimilating all-sky AMSU-A radiance with the assimilation of clear-sky AMSU-A radiance. The study tested a few inflation and localization configurations and used a satisfactory combination. The impact of assimilating all-sky AMSU-A radiance relative to clear-sky AMSU-A radiance is generally consistent with previous studies, showing positive impact on the short-term and long-term forecasts in wind and moisture, and some degradation in temperature in the southern hemisphere. Although all-sky AMSU-A radiance is only assimilated over the water, verifications using radiosondes show that the influence propagates over the land in a few days, showing mostly global improvements in forecast accuracy.

This paper is overall well prepared, constructed, and presented. I have a few concerns, but none of them are major, and this paper should be ready for publication with a handful of minor revisions. My comments are listed below.

1. The novelty of this study is not exactly clear to me. The implementation of LGETKF to MPAS-JEDI was done by a previous work, and a similar study assessing the impact of all-sky AMSU-A radiance using MPAS-JEDI, albeit 3DEnVar instead of EnKF, and did not include clear-sky MHS radiance as in the study, was already performed with similar conclusions. It would be helpful to refine the scope and highlight the novelty of this current study.

Reply: We sincerely appreciate the reviewer's valuable comment. While the LGETKF has been implemented within OOPS (Frolov et al., 2023) for multiple model interfaces in the JEDI framework, this study presents its first implementation and evaluation specifically for MPAS-JEDI. Previous work (Liu et al., 2022) demonstrated MPAS-JEDI's all-sky radiance assimilation using the 3DEnVar method. However, the configurations of satellite radiance assimilation in the LGETKF differs in several aspects—for example, the configuration of localization for all-sky radiances. Even though the overall conclusions may be similar to Liu et al. (2022) and other researches, our work offers a practical reference for configuration and performance of MPAS-JEDI's LGETKF. We have revised the introduction and conclusion and hope the revisions clarify the novelty and contribution of our study.

2. The two sensitivity experiments with different inflation parameters ("AllSky-RTPS" and "AllSky-RTPP") are not an apple-to-apple comparison to the final experiment ("AllSky"). More suitable configurations would be either 1) AllSky-RTPS uses  $\alpha_{RTPS}$ =0.9 and AllSky-RTPP uses  $\alpha_{RTPP}$ =0.5, or 2) AllSky-RTPS uses  $\alpha_{RTPS}$ =1.0 and  $\alpha_{RTPP}$ =0.5, and AllSky-RTPP uses  $\alpha_{RTPS}$ =0.9 and  $\alpha_{RTPS}$ =0.7. These configurations will ensure that there is only **one** parameter different in these two sensitivity experiments compared with the "AllSky" experiment, and the difference should be completely a result of this specific parameter, while the comparison now actually includes influences from both  $\alpha_{RTPS}$  and  $\alpha_{RTPP}$ .

Reply: We agree that changing only one parameter in the inflation sensitivity tests would provide a fairer comparison among the three experiments and better isolate the impacts of RTPP

or RTPS configurations. However, the purpose of these experiments is to identify a configuration that ensures a stable ensemble spread across DA cycles. Accordingly, we referred to previous studies for guidance, including  $\alpha_{RTPS}$ =1.0 from DART's EAKF,  $\alpha_{RTPS}$ =0.8 from MPAS-JEDI's EDA, and the combined inflation approach from the Met Office's EDA system. We found that the combined approach is the most effective for our system. We will explore further refinements in future studies, and relevant discussions have been added in the revision regarding other potential adjustments to the inflation configuration in Lines 323-325.

**Other comments:**

Line 74: Mentioning OOPS is the data assimilation solver component of JEDI might be helpful for the readers who are not familiar with the structure of JEDI.

Reply: the sentence has been revised to "the local volume solvers developed by Frolov et al. (2024), which are part of the Object-Oriented Prediction System (OOPS), the DA solver component of JEDI, have been implemented in MPAS-JEDI.".

Line 86: "...is when..." -> "...is applied when..."

Reply: Fixed.

Line 144: What does excluding thinned observations mean? If it is referring to data thinning, it might be better to say something like "unused observations after data thinning."

Reply: Thanks for your suggestion. We have revised it to "unused observations after data thinning".

Table 2 and Figure 10: It might be helpful to list the frequencies of the channels.

Reply: Thanks for your suggestion, we have added frequencies of each channel in Lines 202-205 in the revision.

Line 207: There are two "use"s.

Reply: the duplicated 'use' has been deleted.

Line 210: "significant CWL content" -- how much?

Reply: pixels with the CLW content exceeding 0 in channels 5 and 6 are filter.

Table 3: For ClrSky and AllSky, the second  $\alpha_{RTPS}$  should likely be  $\alpha_{RTPP}$ .

Reply: Thanks for pointing it out. We have revised Table 3.

Line 279: "model level 50" -- there is no mention of the total number of model levels and how they are distributed.

Reply: The model level number is 55, with the height of model top being 30 km. We gave the details in section 3.1. The model level 50 is approximately 25 km AGL, which is on the stratosphere. For reader's reference, we have added '(~25 km AGL)' behind 'model level 50'.

Figure 2: In my opinion, the x-axis label should be "assimilation time"; "assimilation cycles" should correspond to 1, 2, 3, ... etc.

Reply: Fixed.

Line 361: "NOOA" -> "NOAA".

Reply: Fixed.

Line 445-446: Without a clear statement on the novelty of this study, I'm not sure about the accuracy of this sentence.

Reply: We have revised the introduction and conclusion and hope the revisions clarify the novelty and contribution of our study.

Financial support: The grant number listed here is a NOAA grant number, not a USAF one.

Reply: The grant number is actually a USAF one. We follow the previous MPAS-JEDI publications, Liu et al. (2022), Guerrette et al. (2023), and Jung et al. (2024).

**Reply to CC1:**

Comments on the manuscript titled "All-sky AMSU-A radiance data assimilation using the gain-form of Local Ensemble Transform Kalman Filter within MPAS-JEDI-2.1.0: implementation, tuning, and evaluation."

1. The manuscript identifies temperature forecast degradation, particularly in the southern midlatitudes (50°S–60°S), and attributes it to potential microphysical biases in the MPAS model, referencing similar issues in other systems (e.g., ECMWF's 4DVar). However, the discussion lacks sufficient detail on the specific nature of these biases (e.g., liquid water representation in cold-sector clouds) and their implications for all-sky radiance assimilation. To strengthen this section, the authors should:

Provide a more detailed explanation of the suspected microphysical biases, including specific model parameterizations (e.g., WSM6 microphysics scheme) that may contribute to these issues.

Reply: We appreciate the reviewer's insightful comment regarding potential microphysical biases in the MPAS model and their implications for all-sky radiance assimilation. While we acknowledge that detailed investigation of the specific microphysical processes (e.g., liquid water representation in cold-sector clouds, parameterization in the WSM6 microphysics scheme) could provide valuable insights, such an in-depth analysis is beyond the scope of the present study. In the conclusion section, we have noted this as an important direction for future work, where we plan to conduct targeted experiments to diagnose and mitigate these biases, thereby further refining the all-sky assimilation capability.

2. The manuscript briefly mentions the all-sky observation error model, referencing Liu et al. (2022) for details, but does not adequately discuss its implementation or sensitivity in the context of LGETKF. Given the critical role of observation error modeling in all-sky radiance assimilation, the authors should:

Provide a concise summary of the piecewise linear ramp function used for observation errors, including how it accounts for cloud liquid water path variability.

Reply: We appreciate the reviewer's suggestion. In the revised manuscript, we have added a concise description of the all-sky observation error model. Specifically, for each AMSU-A window channel, the observation error is expressed as a piecewise linear ramp function of the averaged cloud liquid water (CLW) path, computed from both the observed and CRTM-simulated (background) radiances of AMSU-A channels 1 and 2. Observations with CLW below a lower threshold are assigned clear-sky errors, while those above an upper threshold use all-sky errors. For CLW values in between, the error increases linearly from the clear-sky to the all-sky value. This formulation makes the observation error cloud-dependent, thereby accounting for CLW-related variability from both the observations and the background. Related words are added in Lines 253-258.

3. The manuscript highlights efforts to improve the computational efficiency of MPAS-JEDI's LGETKF (e.g., separate job steps, parallel HofX calculations, and quality control preprocessing). However, the evaluation of computational performance is limited to a single table (Table 1) with wall-clock times for one analysis cycle. To provide a more comprehensive assessment:

Include a comparison of computational costs with other EnKF-based systems (e.g., DART's EAKF or ECMWF's EnKF) to contextualize the efficiency gains.

Reply: Thank you for the suggestion. We agree that comparing MPAS-JEDI's LGETKF computational performance with other EnKF-based systems (such as DART's EAKF or ECMWF's EnKF) would be valuable. However, such comparisons are challenging due to differences in system configurations, hardware, and experiment setups, and are therefore beyond the scope of this study. Instead, we have added a discussion comparing run times with and without specific improved analysis procedures, demonstrating a significant reduction in MPAS-JEDI's LGETKF run time. We believe this provides a meaningful assessment of the computational efficiency gains achieved. Related statements are available in Lines 166-169.

4. While the manuscript uses bootstrap resampling to indicate statistical significance in Figures 4–10, the methodology for these tests is not clearly described, and the significance markers (e.g., black circles or triangles) are sparingly applied. This raises questions about the robustness of the reported improvements and degradations. The authors should:

Explicitly describe the bootstrap resampling methodology, including the number of resamples, confidence intervals, and how significance thresholds were determined

Reply: We appreciate the reviewer's comment. In the revised manuscript, we have included the description of the bootstrap resampling methodology when it first appears. Specifically, we now state that each statistic (e.g., RMSE difference) is treated as an independent and identically distributed sample, and resampled 10000 times with replacement. The 95 % confidence intervals are computed using the percentile method (Gilleland et al., 2018), corresponding to the 2.5th and 97.5th percentiles of the bootstrap distribution. We have also clarified the temporal coverage of the verification datasets and the sample sizes used for each figure and the.

5. The manuscript emphasizes the inclusion of hydrometeor variables (Qc, Qi, Qr, Qs, Qg) in the LGETKF analysis, but the evaluation of forecast impacts focuses primarily on U, V, T, and Qv, with limited discussion of hydrometeor fields. Given that all-sky radiance assimilation is particularly relevant for clouds and precipitation, the authors should:

Include a dedicated analysis of the forecast impacts on hydrometeor fields (e.g., cloud liquid water, rainwater) using verification against independent observations (e.g., ATMS window channels or precipitation datasets).

Reply: Thank you for your valuable suggestion. We do include verification results against independent observations from ATMS, which were not assimilated in our experiments. Specifically, we analyzed the ATMS window channels 1–3 (shown in Fig. 10a–c), as these channels are sensitive to hydrometeors and precipitation. Our results demonstrate that the all-sky assimilation of AMSU-A window channel radiances leads to a significant positive impact on hydrometeor and precipitation forecasts, with improvements persisting up to 6 days.

6. Introduction section could benefit from a broader contextualization of recent developments in land-atmosphere coupling and surface-atmospheric feedbacks that influence radiance assimilation outcomes. In particular, studies that incorporate vegetation and land surface properties, such as Leaf Area Index (LAI), have shown value in improving hydrometeorological forecasts and surface—subsurface water dynamics. The authors are encouraged to briefly discuss these complementary efforts and cite relevant recent literature, such as "Assimilation of Sentinel-Based Leaf Area Index for Modeling Surface—Groundwater Interactions in Irrigation Districts", which highlights the role of LAI assimilation in enhancing surface—subsurface coupling and its potential relevance to radiance-based DA strategies.

Reply: Thank you for your insightful suggestion regarding the role of vegetation and land surface properties such as Leaf Area Index (LAI) in improving hydrometeorological forecasts and their potential relevance to radiance assimilation. We agree that these factors are important in land–atmosphere coupling and surface–subsurface interactions in many contexts.

However, in the current study, we focus specifically on the assimilation of AMSU-A window channel radiances over water surfaces, and the radiance simulations do not explicitly incorporate land surface properties such as LAI. Additionally, the simulated clear-sky AMSU-A and MHS radiances used in this study are not influenced by land surface properties such as LAI. Therefore, the impact of vegetation and related land surface processes on the radiance assimilation results is minimal under our current experimental setup.

Given these considerations, we believe that a detailed discussion of LAI assimilation and land surface coupling falls outside the scope of this work. Nevertheless, we appreciate the reviewer highlighting this important area for future investigations where land surface—atmosphere interactions might be explicitly included.

---

## Author Response (AR2)

We sincerely thank the editor and the two anonymous reviewers for their positive feedback and the time they dedicated to reviewing our revised manuscript. We have carefully gone through the manuscript again and made revisions based on your comments. In addition, we have carefully checked the manuscript for typos and corrected them wherever possible. Please refer to our detailed point-by-point responses below and the tracked changes in the revised manuscript for reference.

Topic editor decision: Publish subject to minor revisions (review by editor)

The revised manuscript addressed the reviewer comments and is ready for publication pending the follow minor revisions:

Line 538: Add "separately" between "steps" and "with different jobs"

Reply: Fixed.

Line 539: Remove "two steps"

Reply: Fixed.

Line 756: Change to "Trémolet, Y. and T. Auligné"

Reply: Fixed.

Review #1

I appreciate the authors' efforts to better highlight the novelty of this study and to address all my specific comments. I am especially glad to see the runtime comparisons with and without the offline QC step, and the mention of "reduce obs space" for future users of MPAS-JEDI. I recommend publication after a few very minor issues are addressed (see below).

Reply: We sincerely thank the review for your positive feedback and your time on our revised manuscript.

L72: Should "access" be "assess"?

Reply: Fixed.

L225: The new list of channels and their frequencies is a bit wordy and distracting. Could this be moved into a table?

Reply: We have updated table 2 to include the frequency of each channel.

L255: Pixels with CLW contents exceeding 0 are excluded -> add unit for 0 please

Reply: Fixed.

L378: Change "10000" to "10,000".

Reply: Fixed.

**Review #2**

I would like to thank the authors for their efforts in revising the manuscript and addressing my concerns. The manuscript is in good shape now, and I have no more comments.

Reply: We sincerely thank the review for your positive feedback and your time on our revised manuscript.